# Digital Financial Inclusion Sustainability in Jordanian Context

**Abdalwali Lutfi [1], Manaf Al-Okaily [2,\*] , Malek Hamed Alshirah [3], Ahmad Farhan Alshira'h [4],
Thaer Ahmad Abutaber [2] and Manal Ali Almarashdah [2]**

1    Department of Accounting, College of Business, King Faisal University, Al-Ahsa 31982, Saudi Arabia; aalkhassawneh@kfu.edu.sa
2    School of Business, Jadara University, Irbid 733, Jordan; thaer@jadara.edu.jo (T.A.A.); manal@jadara.edu.jo (M.A.A.)
3    Faculty of Economics and Administrative Sciences, Al al-Bayt University, Al-Mafraq 130040, Jordan; shraamalek@gmail.com
4    Faculty of Administrative and Financial Sciences, Irbid National University, Irbid 2600, Jordan; alshraah.ahmad@yahoo.com
\*    Correspondence: m.alokaily@jadara.edu.jo or m.oqaili@ju.edu.jo

**Abstract:** Digital Financial Inclusion (DFI) refers to efforts to make digital financial services available and affordable to all individuals and institutions, regardless of their net expense or institution size and demographic location. Despite the immense benefits of DFI and DFI-based products and services such as mobile money and payment systems, users' acceptance is thin, limited, and disappointing in some developing countries as Jordan. Consequently, this study has investigated the factors influencing the acceptance of the mobile payment system in the Jordanian context. This study's research model synthesizes the Technology Acceptance Model (TAM) variables and extended the model with perceived financial cost as an independent variable. The research model has been empirically confirmed by fitting the model to data collected from 304 Jordanian citizens using a survey instrument. The data were analysed using Partial Least Squares-Structural Equation Modelling (PLS-SEM). The result has confirmed that behavioural intention to use the m-payment system is significantly and positively influenced by perceived usefulness and perceived financial cost; behavioural intention to use m-payment system was not found to be significantly and positively influenced by perceived ease of use and hence the related hypothesis was not supported. Finally, conclusions and recommendations are further discussed in the last section of this paper.

**Keywords:** financial technology; digital financial; digital payment; financial inclusion; Central Bank of Jordan; financial sustainability; digital economy; TAM; MAT

## 1. Introduction

Financial inclusion, including digital payment systems, is not confined to spreading digital services to a larger population; it also includes offering quality and diversified financial services at a reasonable cost [1]. At the macro level, the benefits of financial inclusion and electronic payment are to enhance financial inclusion, enhance the stability of the financial system, enhance economic development, increase employment rates, increase the financial access for enterprises, especially SMEs, increase transparency to fight money laundering and the financing of terrorism, and reduce poverty levels through cutting down the costs of financial transactions [2,3]. Eventually, developing an advanced electronic payment system is crucial for maintaining the strength and efficiency of the sustainable national payment system, using the advantages mentioned above to attain sustainable, comprehensive development and contribute to supporting Gross Domestic Product (GDP) growth.

Recently, Jordan's interest in developing infrastructure for financial inclusion and the digital banking sector has increased, especially in the mobile payment systems domain. As such, the government objective to expand the diffusion of such systems is to achieve and

enhance financial inclusion in the country via mobile phone technology to increase access to formal banking and payment services in the broader population, especially those who could not access the traditional delivery channels such as ATMs, Net Banking, and bank branches. Accordingly, these systems will bring prosperity and socio-economic benefits to underbanked and unbanked population segments. Despite the strong commitment from the Central Bank of Jordan (CBJ) to deepen financial inclusion in a country where less than a quarter of adults have bank accounts, the intended impact has not yet materialized [4]. In other words, it deepens the Jordanian economy financial inclusion vision to reaffirm the basic tenet that all adults have the right to access loans, savings, payments, and insurance services from formal financial institutions easily and at an affordable cost while maintaining the sustainability of the businesses that provide financial services [1].

In today's globally connected marketplace, there is a very high penetration rate regarding smartphones and social media due to rapid technology advancements [5–7]. This is a great opportunity for many countries worldwide to promote their economic growth and support financial stability by adopting financial inclusion through mobile payment methods [3]. Accordingly, financial inclusion can be defined as the state wherein people can access various financial products and services (payment transactions, savings, credit, remittances, and insurance), which are delivered appropriately, reliably, and sustainably at a reasonable cost that meets their needs. Eventually, financial inclusion strives to remove the barriers that exclude individuals from participating in the financial sector services and using these services to improve their living standards.

Mobile payment (m-payment) systems are defined as a means of sustainable payment. The payer employs mobile communication techniques in conjunction with mobile devices to initiate, authorize, or realize payments [4]. As such, it does not require a hefty deployment cost. In the Jordanian context, Mobile payment, otherwise known as Jordan Mobile Payment (JoMoPay), is a new payment method that emerged from the need of the CBJ to provide better financial services to people who live in rural areas who do not maintain a formal relationship or bank accounts. The JoMoPay system is defined as "a payment method using electronic money whether by the customer's using his account at a bank or by opening an e-wallet account in the company" (p. 1, [8]). In other words, the JoMoPay system refers to payment through mobile technology synchronized with bank accounts or via creating an m-wallet account (for those who do not have a bank account). In this regard, the e-wallets apps in current research are all server-based non-bank e-wallets. In Jordan, there are several server-based e-wallets apps providers such as Orang Money, Mahfazti, and Dinarak. Accordingly, the CBJ has launched the JoMoPay system as a novel payment systems method in recent years to improve financial inclusion in a developing country such as Jordan.

Unfortunately, the financial inclusion aspects of including the mobile payment systems were worse than expected compared to Jordanian society culture and the level of development of financial and banking services; hence, those results were not satisfactory for the CBJ [9]. Despite those technological advancements mentioned above and the availability of a good infrastructure to use JoMoPay system, it is still experiencing a low acceptance rate [10,11]. Recent survey results conducted by [12] to explore the current JoMoPay system status in the Jordanian context showed that only a small amount of the total Jordanian population had used the JoMoPay system [13], showing a low JoMoPay system acceptance in Jordan.

Accordingly, the current study is important for several reasons since the outcome results will provide recommendations and valuable guidelines to increase Jordanians' acceptance toward JoMoPay system usage. Therefore, the problems discussed in the aforementioned highlight those research questions that are required to be examined, these are as follows: Can a modified TAM model be used to determine the acceptance of the JoMoPay system among Jordanian citizens? Is there a direct relationship between predictors (perceived usefulness, perceived financial cost and perceived ease of use) and behavioural intention to use the JoMoPay system?

Concerning research objectives, the study intends to examine the influencing factors on the acceptance JoMoPay system among citizens in Jordan. In addition, this study will investigate the factors influencing the intention to use the JoMoPay system. Consequently, the main research objective is formulated: To determine the factors that lead to the JoMoPay system's acceptance among Jordanian citizens using a modified TAM model. The related objective is as follows: To test the direct relationships between perceived usefulness, perceived financial cost, and perceived ease of use and behavioural intention to use the JoMoPay system.

The current paper covers nine main sections beginning with an introduction, which explain the concept of digital financial inclusion including the mobile payment systems, followed by the problem statement, research questions, research objectives, significance, and motivation of the study. The next sections introduce the literature review, MAT and TAM models, and then present the research hypotheses. Section 5 represents the research methodology and Section 6 introduces data analysis and results. The seventh section discusses the research results and provides the implications of the study. Section 8 focuses on the research limitations and recommendations for future directions. Finally, the last section presents the paper's recapitulations and conclusion, thus completing the whole paper.

## 2. Literature Review

Realizing the significance of enhancing financial inclusion, and in order to develop the national payment system in Jordan to keep pace with best international standards and best practices, the CBJ set up inclusive retail payment systems in order to move from the paper-dominated payment environment to the electronic environment, in a way that attains the CBJ's goals of expanding financial inclusion in Jordan [3]. Additionally, both the CBJ and NPC do support this movement through encouraging modern attainable payment initiatives and exploring the possible opportunities to increase investment in advancing the electronic payment process [2]. The CBJ is also looking to transform government payments from paper-dominated to non-paper payments due to the substantial volume and quantity of transactions, in addition to the fact that most of the beneficiaries are unbanked [1]. Thus, the mobile payment systems will make the improvement process more feasible and achievable.

With respect to the mobile payment systems context, the previous study has also considered another theme concerning key drivers of the user's acceptance of mobile payment. In this regard, Qasim and Abu-Shanab [11] tried a first theoretical contribution based on the Unified Theory of Acceptance and Use of Technology (UTAUT) by suggesting a number of variables (namely, performance expectancy, followed by social influences, trust, network externality) as key predictors of a user's willingness to accept mobile payment. Likewise, an empirical study in the context of Jordan conducted by Al-Okaily et al. [10] found that behavioural intention towards using mobile payment systems is largely shaped by the level of performance expectancy, social influence, price value, security, and privacy.

Moreover, Shaw [14] further investigated the mediating effect of trust on m-wallet acceptance in Canada. Drawing from the TAM model, the results revealed that perceived usefulness is the main factor that influences m-wallet acceptance that is mediated by trust. Using an online survey with random sampling, 284 usable samples were gathered and analyzed with PLS-SEM. The results showed significant associations between perceived usefulness, trust, informal learning, and intention to use m-wallet. In addition, information learning also positively influences perceived usefulness. However, there are no significant effects of perceived ease of use and m-wallet self-efficacy on the intention to use and perceived usefulness, respectively.

### 3. Theoretical Models

*3.1. Technology Acceptance Model (TAM)*

The TAM model was originally proposed by [15] in the information technology (IT) acceptance context. It was introduced as one of the most critical IT acceptance theories in the world [16]. Moreover, the TAM model has been selected by past studies to examine IT acceptance and has been certified to be a predictor of IT acceptance and use [17]. Consequently, the success of IT adoption depends on user acceptance [18]. Recent research has investigated the validity of the TAM in the context of mobile internet [19]. The results of the aforementioned study recommend TAM as a strong model for the acceptance of the m-internet. An expansion to the theoretical model of TAM is highly recommended by [15]. The proposed theoretical framework of the present study included the perceived financial cost as an extension to the TAM model to predict Jordanian citizens' intention to accept m-payment systems.

In short, the TAM model investigation of the user's intention to use a particular IT depends on four main steps [15]. The first step examines the effect of external factors on Perceived Usefulness (PU) and Perceived Ease of Use (PEU) of IT. PU rises from the extent to which users believe that embracing a specific system could improve their performance, while PEU refers to the extent to which users believe that embracing a specific system could be free of physical and mental efforts. The second step is when the PU and PEU affect the user's Attitude Towards Using (ATU) a particular system. In the third step, the PU and ATU determine the usage intention. The last step is making the final decision to accept or reject the use of technology [15,17]. In addition, TAM has been contingent on several developments and additions, including the unified theory of the acceptance and use of technology (UTAUT).

*3.2. Mental Accounting Theory (MAT)*

Digital payment, g, is characterized by uncertainties and risks on the part of payers [20]. Thus, theories that describe payer decisions under uncertainties and risks should shed light on payer choice in the digital payment context. Mental accounting theory (MAT) explains financial decision-making of a payer under the aforementioned circumstances [21]. MAT proposes that individuals weigh beneficial results that are deemed assured more intensely than beneficial results that are considered plausible. It is the effect of confidence which make individuals avoid risks to make any decisions comprising benefits, and that clarifies why individuals are likely to desire an options with a certain but lesser gain. Actually, risk avoidance is considered to be one of the well-known details concerning risky selections involving utilities and gains [22].

As claimed by the MAT principle, an individual analyzes transactions in two stages, the judgment and decision-making stage. In the judgment stage, and to evaluate probable transactions, Thaler [21] suggested three kinds of utility: acquisition, transactions, and whole utility. In acquisition utility, the values of the service received matched (compared) to the expenditure. In transaction utility, an individual considers the perceived advantages of a deal or transaction (based on the differences between the price objectives and the price reference that an individual's expect to pay for the service). Whole utility from a service is the totality of acquisition and transaction utility that represents the perceived total values and utilities derived from applying and using a service such as a digital payment system.

### 4. Research Model

The current research aims to understand the factors that influence Jordanian citizens' intention toward acceptance of m-payment systems services. The research model was based on the TAM model and the rationale of MAT (another related factor such as perceived financial cost) as depicted in Figure 1. In this respect, this part focuses on the discussion of the most significant m-payment systems acceptance predictors. These predictors factors are classified based on the constructs of the TAM model. Consequently, the researchers expect that current research will contribute to the TAM model as a theoretical contribution.

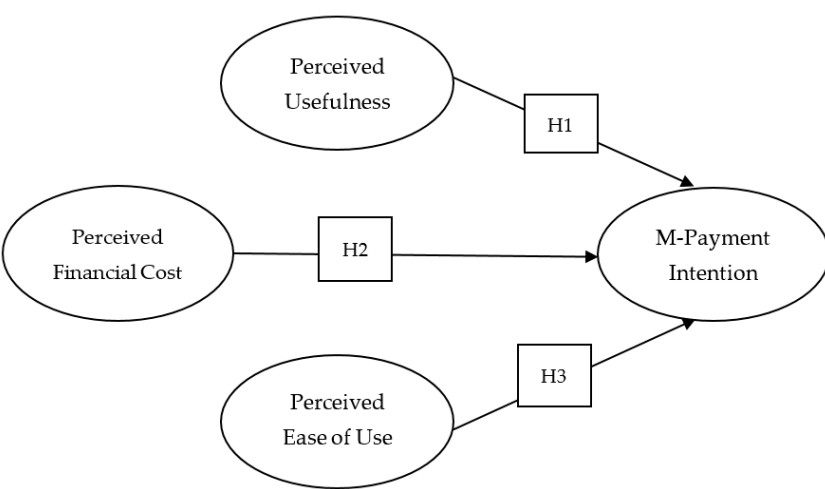

**Figure 1.** Research Model.

*4.1. Perceived Usefulness (PU)*

Perceived usefulness is defined by [15] "the degree to which a person believes that using a particular system would enhance his or her job performance". Perceived usefulness is the same as performance expectancy [23]. Theoretically, PU is identified as the important predictor of the behavioural intention [24–27]. In the context of this study, the formation of PU was to investigate Jordanian citizens' beliefs of whether the mobile payment systems services are useful, then they are more likely to accept these systems. Consequently, we suggest the following hypothesis:

**Hypothesis 1.** *PU will positively impact Jordanian citizens' intention to use m-payment system.*

*4.2. Perceived Financial Cost (PFC)*

Perceived financial cost construct did not come from the TAM model, but literature on IT recommend PFC as an additional construct that might influence the users to accept any new system. PFC is similar to price value in the UTAUT2 model, which can be defined as "the individuals' cognitive trade-off between the perceived benefits of the applications and the monetary cost for using them" [26]. PFC can be seen as monetary sacrifices for applying a service or/and as a sign of service quality. In the context of digital payment, the quality of service is comparable across providers as the services (e.g., digital payment systems) are mostly in nature low-touch, and payers are generally familiar with the service attributes. Therefore, PFC is more frequently measured as monetary sacrifices than a sign of service quality. Based on MAT, PFC affects the monetary dimensions of transactions benefits. Any rise in PFC inferred lesser transaction utilities. As transactions benefits and values are a component of total utility according to MAT, PFC should positively influence total value and utility.

In the current context, numerous studies have found that PFC will be significant for accepting mobile technologies [27,28]. Based on the prior discussion, it is expected that Jordanian citizens are actually interested in the perceived benefits of using mobile payment system services compared to the financial cost paid to use such systems. Hence, the following hypothesis is suggested:

**Hypothesis 2.** *PFC will positively impact Jordanian citizens' intention to use the m-payment system.*

*4.3. Perceived Ease of Use (PEU)*

Perceived ease of use can be defined as the "extent to which users believe that applying a specific system would be free of efforts" [15]. Perceived ease of use is same to effort expectancy [23]. Based on reviewing the previous studies, Perceived ease of use was determined to be a vital factor in the intention of mobile systems [27–31]. In this regard,

it is expected that if the Jordanian citizens' find the mobile payment systems easy to use and effort-free, then they are more likely to adopt these systems. Therefore, this drives the subsequent hypothesis:

**Hypothesis 3.** *PEU will positively impact Jordanian citizens' intention to the use m-payment system.*

## 5. Research Methodology

A quantitative research method was adopted in the current study to test the proposed theoretical framework. In view of this, there are many data collection methods such as mail survey, Internet survey, phone survey, and self-administrated survey [32]. In this study, the self-administered approach is more common in the different contexts to achieve a large response rate, e.g., [33,34]. This is due to some limitations in selecting probability sampling, such as random sampling of Jordanian citizens, for example, the inability to obtain the listing of all Jordanian citizens' names and addresses, citizens scattered all around the country, the high cost of conducting the study in every city, and a hard time accessing certain groups or classes of citizens. On the other hand, the purposive sampling technique is considered fast and easy, where Jordanian citizens can be selected because of their convenient accessibility and proximity to the researchers [35]. Based on the previous argument, the researchers have applied non-probability sampling with the purposive sampling technique.

The Arabic language is the native language of Jordanians. The original measurements were adapted from several previous studies, which were in the English language. However, the use of the Arabic language in the questionnaire of the current study provides ease in understanding for respondents as an advantage in communication. This also enables the researchers to get more insightful information by using the native language of the target population in the data collection process. However, there is little literature published in the Arabic language that investigates a similar context, where survey items are available for the set of variables employed in this study. Therefore, extensive validation and translation procedures have been conducted before the data collection procedures.

In the translation process, since English is not the official language in Jordan, as well as to avoid the impact of cultural and language differences, the researchers followed the back-translation method (forward-backward-translations) suggested by Brislin [36], which is considered to be one of the most popular approaches for questionnaire translation. The services of a professional translation centre were acquired to translate the English version into the standard Arabic language. Later, the Arabic version was retranslated into English by another academic team at the same centre to ensure consistency and avoid mistakes in translation; the original version of the English questionnaire and the translated version were compared. This process is called the back-translation method.

Of the 438 surveys distributed among Jordanian citizens, 352 were completed and 48 were excluded. Of the 304 respondents, 161 (53%) were male and 143 (47%) were female. Only the established variables from prior research have been done, measuring the constructs in 7-Likert Scales. In this regard, the main variables of TAM such as perceived usefulness and perceived ease of use were measured by the same items used by Davis [15]. Meanwhile, perceived financial cost was measured using four items adapted from Venkatesh et al. [23]. Lastly, behavioral intention to use the mobile payment system was measured using four items adapted also from Davis [15], Venkatesh et al. [23]. All items applied are provided into the Table A1 in Appendix A.

## 6. Data Analysis and Results

Researchers used various analytical methods to develop and confirm their research results [37,38]. There are two methods of SEM, covariance-based SEM (CB-SEM) and composite-based SEM (PLS-SEM). It is worth mentioning that the PLS-SEM method has taken up a prominent role within the academic literature of many fields in management science, specifically information systems (IS) research. In general, the PLS model is normally

analysed and interpreted in two stages [39,40]. The first stage, which is the measurement (outer) model, is tested to ensure its validity and reliability. Measurement properties of multi-item constructs, including convergent validity, discriminant validity, and reliability, are examined by conducting Confirmatory Factor Analysis (CFA). The second stage, the structural (inner) model, is analysed by assessing R square, effect size, the predictive relevance of the model, the GoF of the model, and the path coefficient through bootstrapping for testing the hypotheses of the research [41,42]. These two stages are depicted clearly in the following sub-sections.

### 6.1. Measurement Model Assessment

Measurement model assessment is the pre-requisite step for generating results in PLS-SEM. Indeed, Hair et al. [43] described four stages of evaluating the measurement models for PLS-SEM. These stages are (1) indicator reliability is evaluated using indicator loadings of 0.70; (2) internal consistency reliability is evaluated using Composite Reliability (CR) of 0.70 and above; (3) convergent validity is evaluated using Average Variance Extracted (AVE) of 0.50 and above; and (4) discriminant validity is evaluated by using Heterotrait-Monotrait (HTMT) ratio of correlation, Fornell & Larcker criterion, and cross-loading of the indicator. By looking at the cross-loading, the factor loading indicators and the square root of AVE of each latent construct should be higher than its correlation with any other construct in the PLS-SEM model used in a study, with the condition that the cut-off value of factor loading is higher than 0.70 [44–46].

Regarding the measurement model assessment, the outcomes presented in Table 1 clearly support all conventional standards of validity and reliability. Additionally, various statistical analyses reveal that all the measurement items were right and exceeded the recommended value. Subsequently, it can be concluded that the proposed path model has a satisfactory level of validity and reliability. Therefore, the researchers can safely progress toward inner model analysis and test the suggested hypotheses. The next section displays an assessment of the PLS-SEM inner model.

**Table 1.** Assessment of the Measurement Model.

| Variables | Indicators | Reliability | | | | Validity | |
| | | Indicator Reliability | Internal Consistency Reliability | | | Convergent Validity | Discriminant Validity |
| | | Factor Loadings | Cronbach's Alpha | CR | AVE | HTMT |
| | | Loading > 0.70 or >0.40 & Has No Impact on AVE and CR | $\alpha \geq 0.70$ | CR $\geq 0.70$ | AVE $\geq 0.50$ | HTMT < 0.90 |
| Perceived Usefulness | PU1 | 0.954 | 0.942 | 0.962 | 0.895 | Acceptable |
| | PU2 | Dropped | | | | |
| | PU3 | 0.945 | | | | |
| | PU4 | 0.940 | | | | |
| Perceived Financial Cost | PFC1 | 0.936 | 0.941 | 0.958 | 0.850 | Acceptable |
| | PFC2 | 0.948 | | | | |
| | PFC3 | 0.919 | | | | |
| | PFC4 | 0.883 | | | | |
| Perceived Ease of Use | PEU1 | Dropped | 0.921 | 0.944 | 0.849 | Acceptable |
| | PEU2 | 0.926 | | | | |
| | PEU3 | 0.918 | | | | |
| | PEU4 | 0.920 | | | | |

**Table 1.** *Cont.*

| Variables | Indicators | Reliability | | | Validity | |
|---|---|---|---|---|---|---|
| | | Indicator Reliability | Internal Consistency Reliability | | Convergent Validity | Discriminant Validity |
| | | Factor Loadings | Cronbach's Alpha | CR | AVE | HTMT |
| | | Loading > 0.70 or >0.40 & Has No Impact on AVE and CR | $\alpha \geq 0.70$ | $CR \geq 0.70$ | $AVE \geq 0.50$ | HTMT < 0.90 |
| Mobile Payment Intention | MBI1 | 0.927 | 0.957 | 0.969 | 0.885 | Acceptable |
| | MBI2 | 0.957 | | | | |
| | MBI3 | 0.942 | | | | |
| | MBI4 | 0.936 | | | | |

### 6.2. Structural Model Assessment

The structural model was estimated to assess the relationships proposed in the theoretical framework. We employed a bootstrapping approach with 5000 sub-samples to examine those relationships as seen in Figure 1. As a result, the research hypotheses examined by the structural model were all supported except for (Hypothesis H3) as illustrated in Table 2.

**Table 2.** Hypotheses Results.

| No. | Relationship IV→DV | Standard Beta | Standard Error | TV | PV | Sig. | Decision |
|---|---|---|---|---|---|---|---|
| **H1** | PU→MPI | 0.193 | 0.073 | 2.633 | 0.008 | Sig. + | Accepted |
| **H2** | PFC→MPI | 0.108 | 0.057 | 1.911 | 0.056 | Sig. + | Accepted |
| **H3** | PEU→MPI | 0.017 | 0.040 | 0.414 | 0.679 | N.S. | Rejected |

## 7. Discussion and Implications

### 7.1. Discussion

The objective of the present study is to determine factors that influence mobile payment systems usage in the Jordanian context. Thus, in line with that, this study offered new clarifications about the acceptance of digital payment systems using the TAM model in Jordan. To sum up, the major factors that affect the intention of m-payment system acceptance among Jordanian citizens are perceived usefulness and perceived financial cost, while perceived ease of use seems irrelevant. Accordingly, this study provides valuable theoretical implications as discussed below.

### 7.1.1. The Effect of Perceived Usefulness on Intention to Use Mobile Payment System

The empirical results principally show that behavioral intention to use the m-payment system is positively impacted by perceived usefulness. This means that Jordanian citizens are more inclined to increase their use of mobile payment if they recognise that such systems are useful. Consequently, those with high perceived usefulness had high approval for using mobile payment. Jordanian citizens were influenced mostly by their perceptions about the perceived usefulness and the expected advantages of using the mobile payment system. In this regard, this result concurred with the TAM model suggested by Davis [15]. Moreover, this study outcome concurred with existing literature in the mobile payment system area (e.g., [10,11,29,30,47]). Those studies state that increasing perceived usefulness lead to an increased behavioral intention to use those systems.

### 7.1.2. The Effect of Perceived Financial Cost on Intention to Use Mobile Payment System

This study's current results also suggest a positive relationship between perceived financial cost and intention to use mobile payment system. The main reason for this result could be because Jordanian citizens are really concerned about the perceived benefits of using services of mobile payment compared with the cost paid to use these systems. Consequently, the Jordanian citizens are more receptive to being encouraged to use mobile payment systems. In other words, it was showed that Jordanian citizens are most interested in financial issues in formulating their decision to reject or accept the use of the mobile payment system. This result is consistent with other studies, such as [10,19,23,25,48], that perceived financial cost directly impacted using m-payment.

### 7.1.3. The Effect of Perceived Ease of Use on Intention to Use Mobile Payment System

The study results reveal an insignificant relationship between perceived ease of use and behavioural intention to use m-payment system. However, behavioural intention is positively influenced by perceived ease of use as presented in the TAM model [15]. On the other hand, the perceived ease of use does not have a significant effect on this study. Thus, this hypothesis result is not consistent with the prediction of the TAM model. This implies that Jordanian citizens do not place some importance on m-payment systems, although there is a high penetration rate and daily usage of smartphones. Thus, the above reasons suggest that perceived ease of use for this relatively new mobile-based technology is less important [10,11].

### 7.2. Implications

In terms of managerial implication, the indicators that are critical for practitioners to determine to enhance the acceptance of mobile payment systems are perceived usefulness and perceived financial cost. These two indicators are strongly related to the acceptance of mobile payment systems in the Jordanian context. Consequently, the policy-makers in the CBJ need to pay more attention to those indicators by increasing the awareness levels of the advantages of using the mobile payment systems services, as well as confirming transactional safety with its reasonable cost comparing with other systems. Thus, it will lead to improvement of the payment systems and increase the payment options for the people in long term, one of the major objectives of increasing financial inclusion among Jordanian society which the CBJ seeks to achieve digital financial stability in the country. Additionally, Jordanian banks and payment service providers, as well as other related parties, should constantly improve their e-payment systems because of the unprecedented growth rates of e-payment implementation in the breakout of Covid-19 crisis. Services delivered should meet users' anticipations as m-payment systems should be effective and convenient to gain more markets share from cash and increase users' awareness. Likewise, the above-mentioned parties must make sure that m-payment systems, channels and tools are always secure to maintain confidence for the users during such periods. Furthermore, this research is the first reporting the drivers influencing m-payment intention in Jordan, which, in turn, could enhance the understanding for decision makers of the role of these drivers in framing the infrastructure of m-payment in Jordan. Moreover, the results undertaken would extend the developers' knowledge by considering these drivers while developing such systems. M-payment systems developers should emphasise forming m-payment systems that can work on diverse kinds of devices, enhance the performance of payment transactions, achieve tasks in the shortest time frame, are easy to use, user-friendly, and not expensive in order to sustain the user's intention to use m-payment applications in Jordan. More importantly, the bank's decision makers should organize awareness battles to educate users and make them aware of the m-payment risks that could occur throughout the transactions adopting m-payment applications if used incorrectly.

## 8. Limitations and Recommendations

The findings pointed out that the best drivers of m-payment intention is PU, followed by PFC. Given these findings, the current research could offer suitable support for the decision makers in Jordan in building the m-payment infrastructure and make sure that the channels of m-payment are not costly and are capable of executing financial transactions securely and efficiently wherever and whenever the users favor using them.

Regardless of its prominent implications, there are some limitations that need to be highlighted, which could offer fruitful paths for further studies opportunities. As noted, the current study uses the TAM model and MAT combination as a theoretical foundation to conceptualize the influences of proposed drivers on the m-payment system. The findings cannot be justified as the key drivers motivating m-payment systems in general. Therefore, the explanation of findings should be very loose. Accordingly, a suggested model is validated for understanding m-payment systems intention, the current research encourages future studies to verify and replicate the generalization of the model in further countries as well as with other mobile systems in different technology applications (e.g., mobile wallet, mobile banking, mobile learning, mobile health, mobile shopping, mobile government, and mobile digital TV). Consequently, additional studies are definitely required in the technology sector to enhance and verify the validity and applicability of this framework by applying it in different contexts. Finally, this study tested the aforementioned model for studying m-payment (pre-adoption diffusion). Future research may gain a more holistic picture of the post-adoption stage of m-payment at the individual level.

## 9. Conclusions

Generally, the use of mobile phones' applications plays an important role in every aspect of our life, as in the case of mobile payment systems which can be used to save money, time and effort. However, the adoption rate of m-payment systems is still very slow in Jordan, and only a few prior studies have investigated the issues related to it. Despite the weakness of accepting mobile payment system in the Jordanian environment, the ultimate objective to increase the diffusion and acceptance of mobile payment systems is still within the CBJ priorities. Consequently, the current study realized the necessity of considering the factors influencing the Jordanian citizens' intention to use mobile payment systems. This research presumed that three major predictors will determine the intentions to use mobile payment systems; they are perceived usefulness, perceived financial cost and perceived ease of use. Therefore, there were three hypotheses formulated to test the relationship. Results showed that all expected constructs are important and positive influences in predicting intention to use mobile payment systems, except perceived ease of use. Such a result supports all hypotheses except hypothesis number three, as shown in the research model.

**Author Contributions:** Conceptualization, A.L. and M.H.A.; methodology, M.H.A.; software, A.L.; validation, M.H.A. and A.F.A.; formal analysis, M.A.-O.; investigation, A.L.; data curation, A.F.A.; writing—original draft preparation, M.A.-O.; writing—review and editing, T.A.A.; visualization, M.A.A.; supervision, M.A.-O.; project administration, M.A.-O. All authors have read and agreed to the published version of the manuscript.

**Funding:** This research received no external funding.

**Institutional Review Board Statement:** Not applicable.

**Informed Consent Statement:** Not applicable.

**Data Availability Statement:** Not applicable.

**Acknowledgments:** The authors would like to acknowledge the valuable contributions of the reviewers and editor who have provided critical suggestions to improve the quality of the article. The suggested comments have helped in improving the quality of the article considerably. Lastly, we would like to thank the all participants for cooperation and providing pieces of information to contribute to the success of this research.

**Conflicts of Interest:** The authors declare no conflict of interest.

## Appendix A

**Table A1.** Measurements Items and Sources.

| Variables | Code | Measurements Items | Sources |
|---|---|---|---|
| Perceived Usefulness | PU1 | Using the m-payment is useful in my daily life. | Davis [15] |
| | PU2 | Using the m-payment system increases my productivity. | |
| | PU3 | Using the m-payment system saves my time. | |
| | PU4 | Using the m-payment system enhances my efficiency. | |
| Perceived Financial Cost | PFC1 | M-payment system service is reasonably priced. | Venkatesh et al. [23] |
| | PFC2 | M-payment system services are reasonably priced comparing with other systems (e.g., mobile banking systems). | |
| | PFC3 | M-payment system services is a good value for the fees. | |
| | PFC4 | At the current cost, I think m-payment system will provide a reasonable and good cost. | |
| Perceived Ease of Use | PEU1 | I feel that the m-payment system is easy to use. | Davis [15] |
| | PEU2 | I feel that the m-payment system is convenient. | |
| | PEU3 | Getting the information that I want from the m-payment system is easy. | |
| | PEU4 | The m-payment system requires no training. | |
| Mobile Payment Intention | MBI1 | I am planning to use m-payment frequently. | Davis [15], Venkatesh et al. [23] |
| | MBI2 | I expect that I would use m-payment system in the near future | |
| | MBI3 | It would be very likely that I will use m-payment system in the near future | |
| | MBI4 | I seriously intend to use m-payment system in the near future | |

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
