# Peer review of "Digital Financial Inclusion Sustainability in Jordanian Context"

_sustainability, doi:10.3390/su13116312_

Round 1

Reviewer 1 Report

The use of virtual payments has dramatically increase for carrying out transactions Therefore, this article becomes important due to the collection of empirical evidence of this expectancy as well as organizing theories that may contribute to understand users’ adoption intention of mobile payment.

The structure of the article is chosen correctly and it exhibits a logical concern. However, I have a few recommendations about the article:

The title of this article does not reflect its content. I do suggest to redefine title.

In literature review (section 2) would be more complete if the authors combine the Technological Acceptance Model (TAM) with Mental Accounting Theory (MAT) that would allow to integrate technological issues with mental perceptions on technology adoption (Park et al, 2018 and Cheng and Huang, 2013).

To posit the second research hypothesis, authors refer UAUT2 but had not developed the arguments of this Theory. It would be worth to include in the literature review. In addition, the third hypothesis would benefit from the MAT.

Regarding the sample, I was concerned with close to 80% is under 40 years old and more than 80% exhibit a superior Education.

The quantitative data analysis and all tests conducted follow previous literature. While results are consistent with previous literature regarding the H1 and H2 in the case of H3 the result is unexpected. In my opinion authors might use MAT to find a reason to that result.

Park, J., Ahn, J., Thavisay, T., & Ren, T. (2018), “Examining the role of anxiety and social influence in multi-benefits of mobile 711 payment service”, Journal of Retailing and Consumer Services, 47, pp. 140–149.

Cheng, Y. H., & Huang, T. Y. (2013), “High speed rail passengers' mobile ticketing adoption”, Transportation Research Part C: 651 Emerging Technologies, 30, pp. 143–160.

Reviewer 2 Report

Dear Authors,

First of all I want to thank you for the effort to write this manuscript but I have some important considerations that in my point of view are of high importance:

- What is the difference between this article and other articles already published with superior samples and highly cited?

- What is its originality?

- The sample should be extended to have a correct generality.

- The majority of the sample is supposed to be students because of their education and age. is it extensible for generality?

- The TAM model they use is not complete and it is always expected to have at least all the factors of the TAM or UTAUT model and to proceed to an extension of it. Why did they choose the TAM model and not UTAUT or other models in the literature?

- Is it known how many surveys have been discarded for not complying with the established requirements? Has a pre-test been carried out correctly?

- Have indirect effects been taken into account in the PLS model and has a multigroup analysis been taken into account?

- It states in the theoretical implications section that TAM has never been used to analyze Jordanian users. But when we review the theory we can see a significant number of articles already applying it with the same sample:

*Khasawneh, M. H. A. (1970). A mobile banking adoption model in the Jordanian market: an integration of TAM with perceived risks and perceived benefits. The Journal of Internet Banking and Commerce20(3).

* Al Shibly, H. H. (2011). An extended Tam Model to evaluate user's acceptance of electronic cheque clearing systems at Jordanian Commercial Banks. Australian Journal of Basic and Applied Sciences5(5), 147-156.

* Nassar, M. (2013). Mobile devices usage in Jordanian Banking sector: Critical success factors based on an improved technology acceptance model (TAM). International Journal of Sciences: Basic and Applied Research7(1), 93-103

- Last but not least... What does this article have to do with sustainability? Since it is the subject of the journal.

I hope that all these questions and comments can help to improve your manuscript.

Reviewer 3 Report

The paper is interesting and covers current issues related to the acceptance of mobile payment in the Jordanian context, however, there are several possibilities to improve your paper:

  • Introduction: add a brief explanation of why experience from the Jordanian market may be valid and interesting for the international readers
  • introduction and literature review: add a definition of digital financial stability; now it appears only in the title; it is unclear in which way mobile payments are linked to digital financial stability
  • the paper lacks a separate literature review part, which may provide a theoretical framework with regard to the research gap and hypotheses development; literature review should be related to the financial innovation in general (diffusion process and its determinants) and new solutions in the payment system in particular (especially in the context of users' acceptance of new solutions) - it should be added to the paper as a separate subchapter after the introduction
  • add a brief description and data illustrating the situation in the Jordanian payment system (as a broader context for your survey) - how is it organized? what kind of payment instruments dominate? what is the usage of mobile payments? what kind of mobile payment schemes are available?
  • in line 342 you write about "eleven hypotheses formulated", while in the paper only 3 hypotheses are discussed and tested - it should be checked and corrected
  • it may be useful to examine further the link between the particular features of respondents (gender, age, education) and their attitude towards mobile payments (are there any differences among respondents with regard to their attitude and the way they perceive different factors)

Reviewer 4 Report

Dear Authors,

I appreciate your great effort when preparing the article. The topic of digital financial sustainability and factors influencing its development in particular countries is appealing and significant for a broad circle of readers. However, I would suggest you the following improvements:

  1. Please explain why the majority of the participants of the survey are people holding University degrees. I believe the problem of accessibility to financial services and maintaining financial sustainability is more relevant to distant and less 'digitalized' areas where the overall level of education can be lower. Moreover, you point out financial inclusion in the introductory part of the article, and I hope you will develop this idea in the context of University education.
  2. I suggest presenting more descriptive statistics for the sample and responses to particular questions from the survey. It can be shown in the form of tables or charts. It will significantly improve the transparency of the research and helps understand the conclusions in the following parts. I recommend adding a survey instrument in the attachment.
  3. Following the description of the research methods, I would suggest presenting the results for the quantitative model in the form of an equation describing the relations between the variables. It is not apparent how you parameterized the participants' responses and integrated the results into the model. I would recommend extending both the description of the model and adding more details on empirical results.

Round 2

Reviewer 1 Report

The article has been substantially improved, in particular regarding the theoretical background.

Author Response

Dear Reviewer
I hope you are doing well

Kindly find the revised version of our paper, I inserted all the comments as well as corrections of the editor in yellow colour as your request. I would like to take this opportunity to extend my appreciation to you and all reviewers for their insightful comments on the paper, as these comments led me to an improvement of the work. My revisions reflect all reviewers’ suggestions and readers’ comments. Really I am grateful for your assistance as well as your valuable guidance.

Sincerely,
Corresponding author
Dr. Manaf Al-Okaily
Jadara University
m.alokaily@jadara.edu.jo

Reviewer 2 Report

The comments remain the same. Since I rejected the article because of errors that cannot be salvaged, unless the authors again perform a proper sample with proper methodology.

Author Response

(The authors gave the same response as above.)

Reviewer 4 Report

Dear Authors,

Thank you for your comments on the review. The new title of the article seems ambiguous to me. I would suggest changing the title to fit the scope of the problem.

Author Response

(The authors gave the same response as above.)
